# Factors associated with the enrollment of commercial medical insurance in China: Results from China General Social Survey

Songyue Xue[1], Wu Zeng[2], Xiaocong Yang[3], Jianguo Li[1], Lei Zhu[4]*, Guanyang Zou[1]*

1 School of Public Health and Management, Guangzhou University of Chinese Medicine, Guangzhou, China, 2 School of Public Health, Fujian Medical University, Fuzhou, China, 3 School of Public Administration, Guangzhou University, Guangzhou, China, 4 School of Postgraduate Studies, Guangzhou University of Chinese Medicine, Guangzhou, China

☯ These authors contributed equally to this work.
¤ Current address: Department of Global Health, School of Health, Georgetown University, Washington, DC, United States of America
* zhulei@gzucm.edu.cn (LZ); Gzou2023@outlook.com (GZ)

**Data Availability Statement:** The data underlying this study are from the Chinese General Social Survey (CGSS), which is publicly accessible in the project repository. Access to these data is granted upon registration and application at the following

## Abstract

### Background

The Chinese government has been promoting commercial medical insurance (CMI) in recent decades as it plays an increasingly important role in addressing disease burden, health inequities, and other healthcare challenges. However, compared with developed countries, the CMI is still less fledged with low coverage.

### Objective

This study aims to explore the factors associated with enrollment in CMI, with regards to explicit characteristics (including sociodemographic characteristics and family economic status), latent characteristics (including social security status), and the global incentive compatibility index (including health status), to inform the design of CMI to improve its coverage in China.

### Methods

Based on the principal-agent model, we summarized and classified the factors associated with the enrollment in CMI, and then analyzed the data generated from the Chinese General Social Survey in 2015,2018 and 2021 respectively. A comparison of factors regarding sociodemographic characteristics, family economic status, social security status, and health status was conducted between individuals enrolled and unenrolled in CMI using Mann-Whitney U test and Chi-square test. Binary logistic regression analysis was used to explore factors influencing the enrollment status of CMI.

website: http://cgss.ruc.edu.cn/. Additionally, the minimal data used in this study can be found in the Supporting Information files.

**Funding:** GZ acknowledges the financial support from the National Social Science Foundation of China (Grant No. 20&ZD122, http://www.nopss.gov.cn/), the Guangzhou Philosophy and Social Science Development Foundation (Grant No. 2022GZGJ58, https://gzsk.org.cn/index.php?m=content&c=index&a=show&catid=46&id=6237), and the Open Bidding for Selecting the Best Candidates, Guangzhou University of Chinese Medicine (https://www.gzucm.edu.cn/). XY gratefully acknowledges the financial support from the National Natural Science Foundation of China (Grant No. 72104058, https://www.nsfc.gov.cn/), and the Humanities and Social Sciences Foundation of China's Ministry of Education (Grant No. 18YJCZH221, http://www.moe.gov.cn/). The funders had no role in the study design, data collection and analysis, decision to publish, or preparation of the manuscript.

**Competing interests:** The authors have declared that no competing interests exist.

## Results

Of all individuals, the proportion of enrolled individuals shows an increasing trend year by year, with 8.7%,11.8% and 14.1% enrolled in CMI in 2015,2018 and 2021, respectively. The binary regression analysis further suggested that the factors associated with the enrollment in CMI were consistent in 2015,2018 and 2021.We found that individuals divorced, obese, who had a higher level of education, had non-agricultural household registration, perceived themselves as the upper social status, conducted daily exercise, had more family houses, had a car, had investment activities, or did not have basic health insurance were more likely to be enrolled in CMI.

## Conclusions

We identified multidimensional factors associated with the enrollment of CMI, which help inform the government and insurance industry to improve the coverage of CMI.

## Background

Commercial medical insurance (CMI) or private health insurance (PHI) is an important approach to financing health. Worldwide, CMI can be classified into three general categories: free market health insurance, controlled market health insurance, and medical savings accounts (MSAs) [1]. In many, predominantly publicly financed health systems, market-oriented healthcare reforms are being implemented or have been proposed [2]. The health system in the United States is characterized by the dominance of a "market-oriented" medical insurance model using CMI. CMI is used as a supplement or alternative to publicly funded health insurance schemes in many other countries. CMI also plays an important role in the medical safety net system [3, 4], and the coverage is relatively high in many developed countries.

CMI appeared in China in the 1980s. Different from Germany and many other developed countries, where CMI is a substitute for publicly funded health insurance such as social health insurance, CMI in China is often supplementary to public health insurance [5, 6]. At present, China has established a multi-level medical safety net system, where CMI is included as a component. Due to the concern about the increasing disease burden and health inequity among the population, the Chinese government started to promote CMI in recent decades [7]. The share of health expenditure from CMI claims among the total medical expenditure in China increased from 2.51% to 4.04%, from 2017 to 2020, and with low coverage of CMI, there is still room to grow when compared with developed countries.

To encourage the enrollment of CMI and expand the coverage of CMI in China, it is critical to understand the factors affecting the enrollment. To date, most studies on the enrollment of CMI have been focused on demand-side factors. According to the idea of the principal-agent model, the determinants of health insurance enrollment could be divided to three categories, including explicit characteristics(e.g. age, gender, income, employment, education, and family size), latent characteristics which satisfies the necessary conditions of the model (e.g. risk-averse or risk-seeking), and the global incentive compatibility index (e.g. desire for health insurance) [8]. Evidence shows that socio-demographic characteristics and family economic status had a significant impact on the decision to enroll in CMI, including sex [6, 9–11], age [6, 12, 13], marital status [6, 14], household registration and individual mobility [9, 15–17], occupational status [11, 12, 17], education level [6, 11, 12, 14, 16, 18], family size [6, 11, 13, 14, 19], income, and property [6, 11, 15, 17, 19–24]. These factors, which are more related to the ability to pay for

health insurance, exposure to health insurance information, and perceived urgency to health risks, are explicit characteristics factors associated with CMI. The participation in social health insurance can be regarded as latent characteristic factor associated with CMI [9, 25], because the social security status can reflect the psychological characteristics of individuals on risk preference and perception of insurance. This is consistent with some studies about the relationship between psychological factors and the enrollment of CMI [13, 26, 27]. Some unenrolled individuals may benefit from various solidarity mechanisms such as standard social security mechanisms, and the public insurance provided through Medicaid may reduce the willingness to pay for any private insurance mechanism [24, 28]. Moreover, the global incentive compatibility index in the principal-agent model, similar to health-related factors, are more associated with the demand for health services, such as physical health status, lifestyle, past health experiences, and health care spending [19, 20, 24, 29, 30]. Other factors, such as the geographic location of residents, further affect the consumption of CMI. For example, Li et al. found that the maturity of the insurance industry has significantly promoted the purchase of CMI by rural residents [31]. Among the limited studies that examine influencing factors from the supply side of health insurance, Rice et al. found that enrollment decisions might be influenced by the perceived quality of higher-premium plans, better provider networks, lower cost-sharing for services, more generous benefits, and a preference for certain brand-name products [32].

While international literature is available on the factors associated with enrollment of CMI, similar studies remain limited in the context of China. In general, these studies failed to explore the factors associated with the enrollment of CMI in a comprehensive and systematic way for the general population. Prior studies have focused on specific populations such as middle-aged and elderly people, and migrants [10, 18], with a smaller sample size [8, 17]. While the coverage of publicly funded health insurance is on the increase in China, we have observed a subsequent moderate increase in the coverage of CMI. The Chinese government encourages residents to purchase CMI against financial risks from seeking medical care. Understanding factors affecting enrollment in CMI in such a context would help inform where to expand the coverage of CMI to better finance health in a country. Based on the principal-agent theory, this study analyzes potential factors associated with the enrollment of CMI using data generated from a nationally representative household survey in China.

## Methods

### Study design

We applied the principal-agent theory to understand the explicit characteristics, latent characteristics and the global incentive compatibility index associated with the enrollment in CMI in China, using public data generated from the China General Social Survey (CGSS) in 2015, 2018, and 2021. Since the authors used public data for analysis, they had no access to information that could identify individual participants during or after data collection.

### Data sources

The CGSS, a nationwide, comprehensive, and repeated cross-sectional social survey, has been conducted on an annual basis since 2003. The survey had a sample size of more than 10,000 households across the country every year and collects data from communities, families, and individuals. And the data were collected and recorded by trained interviewers during the survey. We used CGSS in 2015, 2018, and 2021 that contained respondents from 29 provinces/cities/autonomous regions in China. After removing missing data, the datasets in 2015, 2018, and 2021 contain 10,305, 11,666, and 7,252 samples, respectively. Data was accessed on March 4, 2022.

### Ethics approval

After ethical approval, the Chinese General Social Survey (CGSS) launched in 2003, is the earliest national representative continuous survey project run by an academic institution in China mainland. We used the public data of CGSS, no additional ethics approval was needed due to the nature of the de-identified administrative dataset.

### Study variables

Based on the principal-agent framework, the independent variables were classified into three dimensions in this study, The explicit characteristics are defined to include sociodemographic characteristics (sex, age, marital status, education, number of family members, current work experience and status, household registration status, subjective social status), and family economic status (family house, car, investment, level of family economy). We take social security status (basic medical insurance, basic pension insurance) as latent characteristics. The global incentive compatibility index include health status (BMI, self-assessed physical health, hospitalization, frequency of physical exercise). The selection of the relevant data, the detailed measurement of the independent variables and how they were coded can be found in S1 File. It should be noted that the question on subjective social status did not appear in the 2015 questionnaire, and it was omitted from the analysis. Additionally, the question on hospitalization only appeared in the 2018 questionnaire, and we obtained the response on this item from 11,666 individuals in 2018 only.

### Data analysis

Data were analyzed using IBM SPSS Statistics, version 22.0. After removing missing data, we analyzed the data for 2015, 2018, and 2021 respectively (See S2 File for the selected data set). Descriptive statistics were generated to report the general characteristics of CMI-enrolled and unenrolled individuals. In the descriptive analysis, mean and median were calculated for continuous variables, while counts and proportions were generated for categorical variables.

   Univariate analyses were performed as a basis for logistic regression analysis. The univariate analyses were used to compare general characteristics between enrolled and unenrolled individuals. Chi-square test was used for categorical variables, and the Mann-Whitney U test was used for continuous variables as the distribution of continuous variables was skewed after the Kolmogorov-Smirnov test.

   To explore the association of potential factors associated with enrollment of CMI, a multivariable logistic regression model was used. The dependent variable was whether to enroll in CMI (Yes = 1, No = 0). The independent variables were explicit characteristics (sociodemographic characteristics and family economic status), latent characteristics (social security status), and the global incentive compatibility index (health status). The independent variables with a p-value less than 0.2 in the Mann-Whitney U test and Chi-square test were included in the subsequent binary logistic regression analysis. The results were presented as adjusted ORs with 95% confidence interval (CI). The significant level was set at 5%.

## Results

### General characteristics of CMI enrollees compared to CMI non-enrollees

Table 1 shows the sociodemographic characteristics in the explicit characteristics dimension of the study sample. Of all individuals, 892 (8.7%), 1,379 (11.8%) and 1,019 (14.1%) enrolled in CMI in 2015,2018, and 2021, respectively. During these three years, the mean age of the sampled individuals was around 50; about 53% were female; more than 70% were married; around

**Table 1. Explicit characteristics: Sociodemographic characteristics of CMI enrolled and unenrolled persons.**

| Variables | All people | | | Enrolled | | | Unenrolled | | | P-value | | |
|---|---|---|---|---|---|---|---|---|---|---|---|---|
| Year | 2015 (n = 10,305) | 2018 (n = 11,666) | 2021 (n = 7,252) | 2015 (n = 892) | 2018 (n = 1,379) | 2021 (n = 1,019) | 2015 (n = 9,413) | 2018 (n = 10,287) | 2021 (n = 6,233) | 2015 | 2018 | 2021 |
| Sex (n, %) | | | | | | | | | | $\chi^2 =$ 5.228/P = 0.022 | $\chi^2 =$ 0.129/P = 0.72 | $\chi^2 =$ 0.071/P = 0.79 |
| Male | 4834(47) | 5484(47) | 3324(46) | 451(51) | 642(47) | 471(46) | 4383(47) | 4842(47) | 2853(46) | | | |
| Female | 5471(53) | 6182(53) | 3928(54) | 441(49) | 737(53) | 548(54) | 5030(53) | 5445(53) | 3380(54) | | | |
| Age (Mean, Median) | 50.56(50) | 52.05(53) | 51.63(53) | 43.88 (43) | 43.53(42) | 43.42(42) | 51.20(51) | 53.19(54) | 52.97(55) | Z = -12.875/ P<0.01 | Z = -20.981/ P<0.01 | Z = -17.084/ P<0.01 |
| Marital status (n, %) | | | | | | | | | | $\chi^2 =$ 68.191/ P<0.01 | $\chi^2 =$ 108.200/ P<0.01 | $\chi^2 =$ 93.455/ P<0.01 |
| Married | 8070(78) | 8917(76) | 5272(73) | 691(77) | 1054(76) | 714(70) | 7379(78) | 7863(76) | 4558(73) | | | |
| Widowed | 950(9) | 1157(10) | 668(9) | 34(4) | 51(4) | 34(3) | 916(10) | 1106(11) | 634(10) | | | |
| Divorced | 212(2) | 299(3) | 216(3) | 18(2) | 43(3) | 48(5) | 194(2) | 256(3) | 168(3) | | | |
| Unmarried | 1073(11) | 1293(11) | 1096(15) | 149(17) | 231(17) | 223(22) | 924(10) | 1062(10) | 873(14) | | | |
| Education (n, %) | | | | | | | | | | $\chi^2 =$ 547.301/ P<0.01 | $\chi^2 =$ 1038.977/ P<0.01 | $\chi^2 =$ 491.436/ P<0.01 |
| Primary school and below | 3917(38) | 4262(37) | 2316(32) | 110(12) | 146(11) | 127(13) | 3807(40) | 4116(40) | 2189(35) | | | |
| Junior high school | 2931(28) | 3184(27) | 2128(29) | 212(24) | 267(19) | 233(23) | 2719(29) | 2917(28) | 1895(31) | | | |
| High school or secondary school | 1840(18) | 2132(18) | 1333(19) | 220(25) | 333(24) | 209(20) | 1620(17) | 1799(18) | 1124(18) | | | |
| Junior college and above | 1617(16) | 2088(18) | 1475(20) | 350(39) | 633(46) | 450(44) | 1267(14) | 1455(14) | 1025(16) | | | |
| Number of family members (Mean, Median) | 3.48(3) | 3.15(3) | 3.44(3) | 3.25(3) | 2.94(3) | 3.40(3) | 3.50(3) | 3.17(3) | 3.45(3) | Z = -3.161/ P = 0.002 | Z = -2.696/ P = 0.007 | Z = -0.078/ P = 0.938 |
| Current work experience and status (n, %) | | | | | | | | | | $\chi^2 =$ 337.305/ P<0.01 | $\chi^2 =$ 570.716/ P<0.01 | $\chi^2 =$ 259.171/ P<0.01 |
| Engage in non-agricultural work | 3775(37) | 4351(37) | 2607(36) | 574(64) | 909(66) | 588(58) | 3201(34) | 3442(33) | 2019(33) | | | |
| Farming | 2137(21) | 2008(17) | 1144(16) | 64(7) | 70(5) | 66(6) | 2073(22) | 1938(19) | 1078(17) | | | |
| Not in employment | 4393(42) | 5307(46) | 3501(48) | 254(29) | 400(29) | 365(36) | 4139(44) | 4907(48) | 3136(50) | | | |
| Household register (n, %) | | | | | | | | | | $\chi^2 =$ 278.206/ P<0.01 | $\chi^2 =$ 317.044/ P<0.01 | $\chi^2 =$ 165.474/ P<0.01 |
| Agricultural household | 5866(57) | 6428(55) | 4332(60) | 272(31) | 451(33) | 422(41) | 5594(59) | 5977(58) | 3910(63) | | | |

(*Continued*)

**Table 1.** (Continued)

| Variables | All people | | | Enrolled | | | Unenrolled | | | P-value | | |
|---|---|---|---|---|---|---|---|---|---|---|---|---|
| Non-agricultural household | 4439(43) | 5238(45) | 2920(40) | 620(69) | 928(67) | 597(59) | 3819(41) | 4310(42) | 2323(37) | | | |
| Subjective social status (n, %) | | | | | | | | | | – | $\chi^2 =$ 135.181/ P<0.01 | $\chi^2 =$ 39.808/ P<0.01 |
| Lower | - | 6479(55) | 3996(55) | - | 598(43) | 472(47) | - | 5881(57) | 3524(56) | | | |
| Middle | - | 4539(39) | 2818(39) | - | 634(46) | 462(45) | - | 3905(38) | 2356(38) | | | |
| Upper | - | 648(6) | 438(6) | - | 147(11) | 85(8) | - | 501(5) | 353(6) | | | |

35% had an education of primary school and below; the median number of family members was 3; more than 52% were employed, of which about 37% were engaged in non-agricultural work; more than 55% of were from agricultural families; 39% perceived themselves as the middle social status.

Compared with non-enrollees, CMI-enrollees had a higher proportion of male (51 vs. 47, p<0.05 in 2015), had a significantly lower median age (43 vs. 51, p<0.05 in 2015;42 vs. 54, p<0.05, 2018; 42 vs. 55, p<0.05,2021), had a higher proportion of unmarried individuals (17% vs.10%, p<0.05 in 2015;17% vs.10%, p<0.05,2018; 22% vs. 14%, p<0.05,2021), and tended to be better educated with a low proportion of individuals who received education at primary school and below (12% vs. 40%, p<0.05 in 2015;11% vs. 40%, p<0.05, 2018; 13% vs. 35%, p<0.05,2021). The two groups of individuals had the same median family members of 3 members per household, but CMI enrollees had a lower mean number of family members (3.25 vs. 3.50, p<0.05 in 2015; 2.94 vs. 3.17, p<0.05,2018). The CMI-enrollees had a higher proportion of individuals engaging in non-agricultural work than non-enrollees (64% vs. 34%, p<0.05 in 2015; 66% vs. 33%, p<0.05,2018; 58% vs. 33%, p<0.05,2021), and had a higher proportion of individuals who perceived themselves in the middle social status (46% vs. 38%, p<0.05 in 2018; 45% vs. 38%, p<0.05,2021). There was no statistical difference in gender distribution between CMI enrollees and non-enrollees in 2018 and 2021 (p>0.05). There was no statistical difference in the distribution of the number of family members between CMI enrollees and non-enrollees in 2021(p>0.05).

Table 2 shows the family economic status in the explicit characteristics dimension of the study sample in the study. During these three years, the mean number of family houses were 1.11, 1.09 and 1.25, respectively. The proportion of individuals who own a car has increased year by year, accounting for 17%, 29%, 43%. About 9% had participated in investment activities; about 51% were at the average family economy level.

Compared with the non-enrollees, the CMI enrollees had a higher proportion of individuals owning cars (38% vs. 15%, p<0.05 in 2015; 55% vs. 26%, p<0.05,2018; 66% vs. 39%, p<0.05,2021) and were more involved in investment activities (28% vs. 7%, p<0.05 in 2015;28% vs. 7%, p<0.05,2018; 30% vs. 7%, p<0.05,2021). The two groups of individuals had the same median houses per household (1 vs. 1, p<0.05 in 2015,2018 and 2021), but the CMI enrollees had a higher mean number of family houses (1.29 vs. 1.09, p<0.05 in 2015; 1.25 vs. 1.07, p<0.05,2018; 1.44 vs. 1.22, p<0.05,2021). Enrollees were generally more affluent, with a higher proportion of individuals with an average level of family economy (59% vs.53%, p<0.05 in 2015; 60% vs.49%, p<0.05,2018; 57% vs. 50%, p<0.05,2021).

Table 3 shows the social security status in latent characteristics dimension of the study sample. During these three years, the proportion of individuals participating in basic medical

**Table 2. Explicit characteristics: Family economic status characteristics of CMI enrolled and unenrolled persons.**

| Variables | All people | | | Enrolled | | | Unenrolled | | | Statistics /P value | | |
|---|---|---|---|---|---|---|---|---|---|---|---|---|
| Year | 2015 (n = 10,305) | 2018 (n = 11,666) | 2021 (n = 7,252) | 2015 (n = 892) | 2018 (n = 1,379) | 2021 (n = 1,019) | 2015 (n = 9,413) | 2018 (n = 10,287) | 2021 (n = 6,233) | 2015 | 2018 | 2021 |
| Number of family houses (Mean, Median) | 1.11(1) | 1.09(1) | 1.25(1) | 1.29(1) | 1.25(1) | 1.44(1) | 1.09(1) | 1.07(1) | 1.22(1) | Z = -9.942/ P<0.01 | Z = -8.482/ P<0.01 | Z = -8.274/ P<0.01 |
| Have a car (n, %) | | | | | | | | | | $\chi^2$ = 318.291/ P<0.01 | $\chi^2$ = 502.872/ P<0.01 | $\chi^2$ = 252.549/ P<0.01 |
| No | 8568(83) | 8261(71) | 4127(57) | 551(62) | 621(45) | 347(34) | 8017(85) | 7640(74) | 3780(61) | | | |
| Yes | 1737(17) | 3405(29) | 3125(43) | 341(38) | 758(55) | 672(66) | 1396(15) | 2647 (26) | 2453(39) | | | |
| Participation of investment activities (n, %) | | | | | | | | | | $\chi^2$ = 451.140/ P<0.01 | $\chi^2$ = 665.599/ P<0.01 | $\chi^2$ = 480.878/ P<0.01 |
| No | 9417(91) | 10579(91) | 6502(90) | 645(72) | 989(72) | 716(70) | 8772(93) | 9590(93) | 5786(93) | | | |
| Yes | 888(9) | 1087(9) | 750(10) | 247(28) | 390(28) | 303(30) | 641(7) | 697(7) | 447(7) | | | |
| Level of family economy (n, %) | | | | | | | | | | $\chi^2$ = 124.883/ P<0.01 | $\chi^2$ = 181.565/ P<0.01 | $\chi^2$ = 104.924/ P<0.01 |
| Average | 5547(54) | 5915(51) | 3687(51) | 526(59) | 822(60) | 581(57) | 5021(53) | 5093(49) | 3106(50) | | | |
| Below average | 3907(38) | 4990(43) | 2977(41) | 221(25) | 388(28) | 294(29) | 3686(39) | 4602(45) | 2683(43) | | | |
| Above average | 851(8) | 761(6) | 588(8) | 145(16) | 169(12) | 144(14) | 706(8) | 592(6) | 444(7) | | | |

insurance has increased, reaching 91%, 92% and 94% respectively. 70%, 75% and 73% had basic pension insurance in 2015, 2018 and 2021.

Compared with the non-enrollees, the CMI enrollees had a lower proportion of individuals participating in basic medical insurance in 2015 (89% vs. 92%, p<0.05); a higher proportion of individuals participating in basic pension insurance in 2015 (75% vs. 69%, p<0.05). There was no statistically significant difference in the participation in basic medical insurance between CMI enrollees and non-enrollees in 2021 (p>0.05).

**Table 3. Latent characteristics: Social security status characteristics of CMI enrolled and unenrolled person.**

| Variables | All people | | | Enrolled | | | Unenrolled | | | Statistics /P value | | |
|---|---|---|---|---|---|---|---|---|---|---|---|---|
| Year | 2015 (n = 10,305) | 2018 (n = 11,666) | 2021 (n = 7,252) | 2015 (n = 892) | 2018 (n = 1,379) | 2021 (n = 1,019) | 2015 (n = 9,413) | 2018 (n = 10,287) | 2021 (n = 6,233) | 2015 | 2018 | 2021 |
| Participate in basic medical insurance (n, %) | | | | | | | | | | $\chi^2$ = 6.79/ P = 0.009 | $\chi^2$ = 3.386/ P = 0.066 | $\chi^2$ = 0.968/ P = 0.325 |
| Yes | 9404(91) | 10770(92) | 6851(94) | 793(89) | 1256(91) | 956(94) | 8611(92) | 9514(93) | 5895(95) | | | |
| No | 901(9) | 896(8) | 401(6) | 99(11) | 123(9) | 63(6) | 802(8) | 773(7) | 338(5) | | | |
| Participate in basic pension insurance (n, %) | | | | | | | | | | $\chi^2$ = 10.618/ P = 0.001 | $\chi^2$ = 2.708/ P = 0.100 | $\chi^2$ = 1.802/ P = 0.179 |
| Yes | 7189(70) | 8766(75) | 5283(73) | 665(75) | 1061(77) | 760(75) | 6524(69) | 7705(75) | 4523(73) | | | |
| No | 3116(30) | 2900(25) | 1969(27) | 227(25) | 318(23) | 259(25) | 2889(31) | 2582(25) | 1710(27) | | | |

**Table 4. The global incentive compatibility index: Health status characteristics of CMI enrolled and unenrolled persons.**

| Variables | All people | | | Enrolled | | | Unenrolled | | | Statistics /P-value | | |
|---|---|---|---|---|---|---|---|---|---|---|---|---|
| Year | 2015 (n = 10,305) | 2018 (n = 11,666) | 2021 (n = 7,252) | 2015 (n = 892) | 2018 (n = 1,379) | 2021 (n = 1,019) | 2015 (n = 9,413) | 2018 (n = 10,287) | 2021 (n = 6,233) | 2015 | 2018 | 2021 |
| BMI (n, %) | | | | | | | | | | $\chi^2$ = 7.927/ P = 0.019 | $\chi^2$ = 1.166/ P = 0.558 | $\chi^2$ = 3.929/ P = 0.14 |
| Normal range | 6030(59) | 6458(56) | 3869(53) | 516(58) | 759(55) | 545(53) | 5514(59) | 5699(56) | 3324(53) | | | |
| Above normal range | 3229(31) | 4125(35) | 2789(39) | 306(34) | 501(36) | 406(40) | 2923(31) | 3624(35) | 2383(38) | | | |
| Below normal range | 1046(10) | 1083(9) | 594(8) | 70(8) | 119(9) | 68(7) | 976(10) | 964(9) | 526(9) | | | |
| Self-assessment of physical health (n, %) | | | | | | | | | | $\chi^2$ = 78.057/ P<0.01 | $\chi^2$ = 146.299/ P<0.01 | $\chi^2$ = 89.704/ P<0.01 |
| Healthy | 6221(60) | 6752(58) | 3900(54) | 651(73) | 981(71) | 661(65) | 5570(59) | 5771(56) | 3239(52) | | | |
| Average | 2230(22) | 2651(23) | 2034(28) | 164(18) | 280(20) | 272(27) | 2066(22) | 2371(23) | 1762(28) | | | |
| Unhealthy | 1854(18) | 2263(19) | 1318(18) | 77(9) | 118(9) | 86(8) | 1777(19) | 2145(21) | 1232(20) | | | |
| Has been hospitalized due to illness or injury in the past year (n, %) | | | | | | | | | | – | $\chi^2$ = 60.556/ P<0.01 | – |
| No | - | 9425(81) | - | - | 1221(88) | - | - | 8204(80) | - | | | |
| Yes | - | 2241(19) | - | - | 158(12) | - | - | 2083(20) | - | | | |
| Frequency of physical exercise (n, %) | | | | | | | | | | $\chi^2$ = 212.851/ P<0.01 | $\chi^2$ = 359.146/ P<0.01 | $\chi^2$ = 139.012/ P<0.01 |
| Never | 4374(43) | 5022(43) | 2516(35) | 173(19) | 276(20) | 194(19) | 4201(44) | 4746(46) | 2322(37) | | | |
| Sometimes | 4260(41) | 4487(39) | 2980(41) | 523(59) | 801(58) | 557(55) | 3737(40) | 3686(36) | 2423(39) | | | |
| Every day | 1671(16) | 2157(18) | 1756(24) | 196(22) | 302(22) | 268(26) | 1475(16) | 1855(18) | 1488(24) | | | |

Table 4 presents the global incentive compatibility index dimension, including health measurements and health behaviors among individuals. During these three years, the proportion of individuals with BMI within the normal range decreased, with proportions of 59%, 56%, and 53%, respectively. The proportion of individuals who rated themselves as healthy also declined, at 60%, 58%, and 54%, respectively. 19% had been hospitalized in the past year due to injuries or other illnesses in 2018; 43% had never done physical exercise in 2015 and 2018, while in 2021, only 35% had never done physical exercise.

Compared with non-enrollees, the CMI enrollees tended to be healthier, with a higher proportion of individuals who rated themselves healthy (73% vs. 59%, p<0.05 in 2015; 71% vs. 56%, p<0.05, 2018; 65% vs. 52%, p<0.05, 2021); a lower proportion of individuals with BMI within the normal range in 2015 (58% vs. 59%, p<0.05); a lower proportion of individuals who were hospitalized in the past year in 2018(12% vs. 20%, p<0.05), and a lower proportion of individuals who had never done physical exercise (19%vs. 44%, p<0.05 in 2015; 20%vs. 46%, p<0.05,2018; 19%vs. 37%, p<0.05,2021). No significant difference in BMI distribution was found between CMI enrollees and non-enrollees in 2018(p>0.05).

## Comparison of factors associated with CMI in 2015, 2018 and 2021

We found that the factors associated with CMI were consistent in 2015,2018 and 2021.

In terms of explicit characteristics, individuals with a higher level of education were associated with a greater likelihood of CMI enrollment compared to those with lower education levels (OR = 2.33, 95% CI: 1.72–3.16 in 2015; OR = 3.00, 95% CI: 2.34–3.85 in 2018; OR = 2.01, 95% CI: 1.51–2.69 in 2021). Individuals from non-agricultural households were more likely to enroll in CMI compared to individuals from agricultural households (OR = 1.60, 95% CI: 1.32–1.93 in 2015; OR = 1.34, 95% CI: 1.15–1.56 in 2018; OR = 1.37, 95% CI: 1.16–1.62 in 2021). Individuals who perceived themselves as the upper social status were 1.60 times more likely to enroll in CMI compared to those who perceived themselves as the lower social status (OR = 1.60, 95% CI: 1.23–2.09 in 2018). Divorced individuals were 1.48 times more likely to enroll in CMI than married individuals (OR = 1.48, 95% CI: 1.03–2.13 in 2021). However, individuals who were older (OR = 0.98, 95% CI: 0.98–0.99 in 2018 and 2021), unmarried (OR = 0.72, 95% CI: 0.59–0.88 in 2018; OR = 0.78, 95% CI: 0.62–0.99 in 2021), had more family members (OR = 0.95, 95%CI: 0.91–0.99 in 2018) were less likely to enroll in CMI than those who were younger, married, and fewer family members. Compared with individuals engaged in non-agricultural work, farmers were less likely to enroll in CMI (OR = 0.59, 95% CI: 0.44–0.80 in 2015; OR = 0.56, 95% CI: 0.42–0.74 in 2018; OR = 0.68, 95% CI: 0.50–0.92 in 2021), and those who were not employed were also less likely to enroll in CMI (OR = 0.56, 95% CI: 0.47–0.68 in 2015; OR = 0.60, 95% CI: 0.51–0.70 in 2018; OR = 0.75, 95% CI: 0.63–0.88 in 2021). Individuals who had more family houses were more likely to enroll in CMI compared to those with fewer or no family houses, respectively (OR = 1.23, 95% CI: 1.12–1.36 in 2015; OR = 1.10, 95% CI: 1.02–1.20 in 2018). Individuals who had a car were about 1.6 times more likely to enroll in CMI than those with no car (OR = 1.60, 95% CI: 1.35–1.90 in 2015; OR = 1.54, 95% CI: 1.35–1.77 in 2018; OR = 1.64, 95% CI: 1.40–1.92 in 2021). And individuals who had investment activities were more than twice as likely to enroll in CMI as those with no investment (OR = 2.21, 95% CI: 1.83–2.68 in 2015; OR = 2.14, 95% CI: 1.82–2.51 in 2018; OR = 2.64, 95% CI: 2.20–3.18 in 2021).

From the perspective of latent characteristics, individuals without basic medical insurance were nearly twice as likely to participate in CMI as those with basic medical insurance (OR = 1.86, 95% CI: 1.44–2.41 in 2015; OR = 1.68, 95% CI: 1.32–2.12 in 2018). Individuals without basic pension insurance were less likely to enroll in CMI compared to those with basic pension insurance (OR = 0.80, 95% CI:0.66–0.97 in 2015).

In perspective to the global incentive compatibility index, obese individuals were 1.20 times more likely to enroll in CMI than those BMI within normal range according to the BMI range (OR = 1.20, 95% CI: 1.03–1.39 in 2021). Physical exercise was positively associated with CMI. Individuals who conducted daily exercise were more likely to enroll in CMI compared to those who had never done so (OR = 1.79, 95% CI: 1.42–2.27 in 2015; OR = 1.71, 95% CI: 1.42–2.07 in 2018; OR = 1.31, 95% CI: 1.06–1.63 in 2021). (Table 5).

## Discussion

In this study, we found that of all individuals, the proportion of enrolled individuals shows an increasing trend year by year, with 8.7%,11.8% and 14.1% enrolled in CMI in 2015,2018 and 2021, respectively. This is also a significant increase compared to the 6.9% coverage rate estimated from the China's National Health Service Survey in 2013 [33]. However, compared with the United States and some developed European countries, the coverage of CMI in China is relatively low. The differences in coverage may be influenced by factors such as the start time of the development of CMI and policy regulations. For instance, in the United States where the "market-oriented" health insurance model is dominant, 55.3% of individuals were enrolled in CMI in 2016 [34]. The high coverage may be related to the earlier establishment and

**Table 5. Comparison of factors associated with CMI in 2015, 2018 and 2021.**

| Variables | OR (95% CI) | | |
|---|---|---|---|
| Year | 2015 | 2018 | 2021 |
| **Explicit characteristics** | | | |
| Age | – | 0.98 (0.98–0.99) *** | 0.98(0.98–0.99)*** |
| Marital status | | | |
| Married | – | 1 (ref) | 1 (ref) |
| Widowed | – | 0.97(0.70–1.33) | 0.79(0.54–1.15) |
| Divorced | – | 1.16(0.82–1.66) | 1.48(1.03–2.13)* |
| Unmarried | – | 0.72(0.59–0.88) ** | 0.78(0.62–0.99)* |
| Education | | | |
| Primary school and below | 1 (ref) | 1 (ref) | 1 (ref) |
| Junior high school | 1.65(1.28–2.14)*** | 1.52 (1.22–1.89) *** | 1.31(1.03–1.66)* |
| high school or secondary school | 1.98(1.50–2.62)*** | 2.22 (1.76–2.79) *** | 1.37(1.05–1.80)* |
| Junior college and above | 2.33(1.72–3.16)*** | 3.00 (2.34–3.85) *** | 2.01(1.51–2.69)*** |
| Number of family members | – | 0.95 (0.91–0.99) * | – |
| Current work experience and status | | | |
| Engage in non-agricultural work | 1 (ref) | 1 (ref) | 1 (ref) |
| Farming | 0.59(0.44–0.80)** | 0.56 (0.42–0.74) *** | 0.68(0.50–0.92)* |
| Not in Employment | 0.56(0.47–0.68)*** | 0.60 (0.51–0.70) *** | 0.75(0.63–0.88)** |
| Household registration status | | | |
| Agricultural household | 1 (ref) | 1 (ref) | 1 (ref) |
| Non-agricultural household | 1.60(1.32–1.93)*** | 1.34 (1.15–1.56) *** | 1.37(1.16–1.62)*** |
| Subjective social status | | | |
| lower | – | 1 (ref) | – |
| middle | – | 1.09 (0.94–1.26) | – |
| upper | – | 1.60 (1.23,2.09) *** | – |
| Number of family houses | 1.23(1.12–1.36)*** | 1.10(1.02–1.20) * | – |
| Have a car | | | |
| No | 1 (ref) | 1 (ref) | 1 (ref) |
| Yes | 1.60(1.35–1.90)*** | 1.54 (1.35–1.77) *** | 1.64(1.40–1.92)*** |
| Participation of investment activities | | | |
| No | 1 (ref) | 1 (ref) | 1 (ref) |
| Yes | 2.21(1.83–2.68)*** | 2.14(1.82–2.51) *** | 2.64(2.20–3.18)*** |
| **Latent characteristics** | | | |
| Participate in basic medical insurance | | | |
| Yes | 1 (ref) | 1 (ref) | – |
| No | 1.86(1.44–2.41)*** | 1.68(1.32–2.12) *** | – |
| Participate in basic pension insurance | | | |
| Yes | 1 (ref) | – | – |
| No | 0.80(0.66–0.97)* | – | – |
| **The global incentive com compatibility index** | | | |
| BMI | | | |
| Normal range | – | – | 1 (ref) |
| Above normal range | – | – | 1.20(1.03–1.39)* |
| Below normal range | – | – | 0.78(0.58–1.04) |
| Physical exercise | | | |
| Never | 1 (ref) | 1 (ref) | 1 (ref) |
| Sometimes | 1.61(1.33–1.96)*** | 1.53 (1.30–1.80) *** | 1.13(0.93–1.38) |

(*Continued*)

**Table 5.** (Continued)

| Variables | OR (95% CI) | | |
|---|---|---|---|
| Year | 2015 | 2018 | 2021 |
| Every day | 1.79(1.42–2.27)*** | 1.71 (1.42–2.07) *** | 1.31(1.06–1.63)* |

OR odds ratio
* p<0.05
** p<0.01
*** p<0.001.

development of the CMI system in the United States which began around 1945 as a pillar of its medical security system [35], and the strong legal protection for those residents who do not have employer-provided health insurance and do not qualify for public insurance to purchase CMI [36]. Besides, the governments of England, Germany, and the United States have all introduced tax support policies for CMI, which has a positive impact on increasing the CMI coverage. CMI appeared in China in the 1980s. In recent decades, the government began to pay more attention to CMI, which gives great impetus to the CMI industry, and the awareness of protection against financial risks brought by medical problems among Chinese residents has been greatly improved. However, the ability of the CMI to protect against the financial risks of illness remains weak, and individuals mainly rely on basic health insurance and/or out-of-pocket spending to pay for medical expenditures. Thus, the willingness to enroll in CMI remains limited. Besides, supply problems such as high prices, poor claims service experience, and the homogeneity of CMI products, further restrain the growth of CMI enrollment. Therefore, to improve CMI coverage, the government should enhance its support for CMI in terms of policy design, medical data platform, and media publicity; the insurance industry and insurance companies should carry out supply-side reforms and actively use technologies to optimize health insurance schemes that meet population needs and to provide efficient and effectiveness services for CMI clients. The conceptual framework based on the principal-agent model is helpful to understand the systematic factors related to the enrolment of CMI in China.

## Explicit characteristics associated with the enrollment of CMI

Similar to previous studies [11, 37], this study found that elder people were less likely to enroll in CMI. Although the elderly are prone to diseases and are high-risk groups for many diseases because of the aging of organs and bodily function, with a greater demand for medical services, there are several reasons for the findings we found in this study. From the supply side, it may be because insurance companies have age restrictions on the purchase of CMI, requiring elderly to have a medical examination before purchasing to ensure good health and meet strict purchasing conditions. Many insurance companies may refuse to provide the elderly with insurance for fear of financial losses. From the demand side, an increasing proportion of individuals with health risks believe that insurers would not be willing to offer them supplementary insurance contracts [38], and thus they may be discouraged from enrolling in CMI. Additionally, the elderly are not as financially able as the young population and do not know how to avoid risks through insurance due to limited knowledge about CMI. The combination of multiple factors mentioned above results in a lower CMI coverage among the older population.

Consistent with previous study [6], we found that unmarried individuals were less likely to enroll in CMI than those who were married. This may be because married individuals tend to be more responsible, as they need to consider several factors related to work and family and

are under great stress when faced with financial risks brought by medical problems. Different from married people who have stable financial sources, unmarried people generally have a lower level of income and savings, and thus are more reluctant to purchase additional commercial insurance. However, we also found that divorced individuals were more likely to enroll in CMI than those who were married, which is different from previous study [11]. This may be because divorced individuals who have been through marriage are more convinced of the importance of medical insurance.

Similar to previous studies [11, 19, 39], we also found that individuals with more family members were less willing to participate in CMI. As households get larger, the total spending is higher to buy CMI. Furthermore, with the expansion of family size, there are more people to share the economic risks brought by medical problems, especially in the families with more young adults, which results in reduced willingness to enroll in CMI. In addition, with the thought of "more children, more happiness", families with a larger population believe that the more people there are, the more secure they are. This would give these families enough confidence to overcome potential financial risks brought by medical problems, so they have less interest in enrolling in CMI.

This study showed that a higher level of education was associated with a greater willingness to enroll in CMI. Similarly, many studies found that education level is positively associated with the enrollment of CMI [6, 11, 12, 14, 16, 18]. This may be because the higher level of education is directly associated with the capacity to accumulate and understand health-related information in deciding to enroll in CMI. On the other hand, individuals with less education are more likely to earn less, which affects their ability to participate in CMI, while individuals with higher education tend to have decent and formal jobs and incomes, to support the payment of CMI.

Consistent with previous studies [9, 11, 12, 15], this study showed that individuals from non-agricultural households, and those engaged in non-agricultural work, had a higher likelihood of being insured. Compared with those engaged in non-agricultural work, those unemployed and those engaged in farming tend to be in the low-middle income groups, and even the vulnerable groups, which compromises their ability to pay for CMI. On the contrary, non-agricultural workers may receive employer sponsored CMI as an employment benefit. In addition, the fact that individuals from agricultural households are less likely to enroll in CMI may be related to the lower per capita disposable income of residents, the higher incidence of poverty, and the lower educational level of households in rural areas.

Our research found that individuals with cars, investment activities, and more family houses were more likely to enroll in CMI. Similarly, many studies have reported that those in higher-income households and having a higher socioeconomic status could have a positive effect on the enrollment of CMI [11, 15, 17, 19–23, 39]. Consistent with a previous study [40], we also found that individuals who perceived themselves as the upper social status were more likely to enroll in CMI. On the one hand, individuals with higher economic status have sufficient financial conditions to enroll in CMI. These people generally have more medical knowledge and a stronger awareness of risks and express concern about unnecessary health damage or asset losses due to catastrophic diseases. On the other hand, these groups belong to the high-income group and are willing and able to obtain additional and high-quality medical services through enrollment in CMI to obtain a better medical treatment experience. Besides, individuals who perceived themselves as the upper social status have more confidence and better economic conditions to afford the cost of CMI. In addition, individuals who perceived themselves as the upper social status have higher requirements and demands for the quality of medical services when they feel unhealthy [41, 42]. CMI can better meet such high-level medical services through additional services.

## Latent characteristics associated with enrollment of CMI

Like previous studies [9, 25], our study shows that individuals without basic health insurance are more likely to enroll in CMI than those with basic health insurance. This result suggests the crowd-out effect of basic medical insurance on CMI [43]. The improved coverage, benefit package and reimbursement rate of basic medical insurance reduce the demand of the participants with the basic medical insurance for medical services outside the scope of basic medical insurance. However, we found that individuals without basic pension insurance were less likely to enroll in CMI compared to those with basic pension insurance. This may be because individuals without basic pension insurance do not pay attention to the role of insurance, and will not spend more money to enroll in CMI.

## The global incentive compatibility index associated with enrollment of CMI

Compared with those who never exercise, individuals who exercise occasionally or daily are more likely to enroll in CMI. This finding is consistent with the finding of Banks E et al. [44]. This may be because people who have frequent physical exercise pay more attention to their health and have a higher level of awareness of avoiding financial risks brought by medical problems. On the contrary, those who never exercise may have a high-risk preference, and they tend to believe that they do not need to enroll in CMI since they are less likely to maintain good health [45]. We also found that obese individuals were more likely to enroll in CMI than those BMI within normal range. As obese individuals have a much higher risk of disease than normal individuals, it may be that obese individuals feel they need CMI to reduce the financial risk of medical treatment.

A few studies focused on factors affecting the enrollment of CMI, such as sex [6, 9–11], hospitalization [14, 30, 46], medical bills [6, 13, 19, 47]. However, we found that these factors had no significant influence on the enrollment of CMI in the present study.

## Policy implications

Based on the analysis of the factors derived from the principal model, a few policy implications could be drawn from this study to inform the reform and growth of CMI in China. Firstly, in the case of older individuals, the government should consider introducing CMI-subsidy policies for the elderly, while the insurance industry should expand publicity channels and methods to improve the awareness of CMI among the elderly. It is also necessary to scale up social business integrated medical insurance, a kind of inclusive CMI that is more friendly to the elderly, with the advantages of low cost, simple terms, broad coverage, and easy access [48, 49].

Secondly, insurance companies should consider the needs of different family structures when designing CMI schemes tailored to different populations. For example, for families with many members, a "family sharing deductible" scheme can be designed to reduce the deductible and cover as many medical expenses as possible. For unmarried people, "long and short insurance combination" should be designed, that is, when the economic ability is insufficient, the CMI schemes with the short term can be selected. With the increase in income, the insurance premium could be gradually increased for long-term CMI. Meanwhile, tailored methods should be adopted according to individual needs to encourage them to enroll in CMI.

Thirdly, a lower level of education may restrict understanding or overwhelm an individual, which could lead to "omission bias" whereby one prefers the status quo to making a hard decision [50]. Therefore, the government should strengthen health literacy of CMI and raise awareness of risk aversion given that many individuals with lower education lack

understanding of CMI and have poor awareness of risk aversion, resulting in the poor willingness to purchase CMI.

At a time when China has made major historic achievements in poverty alleviation, great attention should be paid to agricultural-related social security policies. To narrow the gap between urban and rural areas, the government should implement preferential policies for farmers to enroll in CMI. It is also necessary to promote rural revitalization, accelerate the growth of a new rural economy to raise farmers' income, and lay an economic foundation for farmers to enroll in CMI. Considering the accessibility problem, more CMI institutions can be set up in rural areas to facilitate the promotion of CMI.

For groups with poor family economic status and who perceive themselves with lower social status, the government can consider increasing the subsidy for them to enroll in CMI. Insurance companies should consider designing differentiated CMI schemes to meet the needs of populations with different economic conditions. At the same time, more CMI schemes with inclusiveness can be launched to provide more affordable medical insurance options.

Insurance companies can add value-added services when designing the package of CMI, services such as medical green channels to attract those who are already enrolled in publicly funded health insurance. Meanwhile, it is necessary to highlight the complementary advantages of CMI, such as covering medical expenses not covered by basic health insurance.

## Strength and limitation

The main strength of this study lies in the large national representative sample and the use of multi-year data (2015,2018 and 2021), which helps improve the validity of our conclusions. In addition, we adapt a conceptual framework based on the principal-agent model, which provides theoretical basis and guidance for the whole study. Under the guidance of the conceptual framework, the factors were divided into three dimensions: explicit characteristics (sociodemographic characteristics, and family economic status), latent characteristics (social security status), and the global incentive compatibility index (health status). It would help develop targeted policy suggestions. However, our study is also subject to a few limitations. First, we removed some variables with missing values or logic problems. However, the proportion of the missing data was small. In 2015, 2018 and 2021, only 2%, 2.3%, 2.3% of the surveyed participants did not have enrollment records, while 4%,6.6%,8.9% of the included samples had missing data or logical problems, respectively. The potential bias due to missing data is limited. Second, while many studies have shown that psychological characteristics play an important role in the enrollment of CMI [13, 26, 27], we could not include such variables in the study as CGSS did not contain them. But the social security status in our study can approximately reflect the psychological characteristics on risk preference and perception of insurance. These details and complexities will be an interesting area for future research. Despite these limitations, our study provides useful information for the government to formulate relevant policies and for CMI companies to better design and promote their insurance packages.

## Conclusion

Our study found that individuals enrolled in CMI in 2015,2018 and 2021 was 8.7%, 11.8% and 14.1%, respectively. Although CMI coverage is rising, it still needs to be improved compared with developed countries. Promoting CMIs could target individuals who are older, unmarried, with low educational level, agricultural household registration, farmers, no house, no car, no investment, low subjective social status, with basic medical insurance, without basic pension insurance, and have no physical exercise.

## Supporting information

**S1 File. The selection of the relevant data, the detailed measurement of the independent variables and how they were coded.**
(DOCX)

**S2 File. Study's minimal underlying data set.**
(ZIP)

## Acknowledgments

The authors would like to acknowledge the Chinese General Social Survey (CGSS) team for collecting nationally representative data, and for making the data public.

## Author Contributions

**Data curation:** Songyue Xue.

**Funding acquisition:** Xiaocong Yang, Guanyang Zou.

**Methodology:** Wu Zeng.

**Project administration:** Guanyang Zou.

**Resources:** Guanyang Zou.

**Supervision:** Lei Zhu, Guanyang Zou.

**Validation:** Wu Zeng, Xiaocong Yang.

**Visualization:** Songyue Xue.

**Writing – original draft:** Songyue Xue.

**Writing – review & editing:** Songyue Xue, Wu Zeng, Xiaocong Yang, Jianguo Li, Lei Zhu, Guanyang Zou.

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
