## [Decision Letter · Decision Letter 0]

2 Oct 2023

PONE-D-23-25788Factors associated with the enrollment of commercial medical insurance in China: results from China General Social SurveyPLOS ONE

Dear Dr. Zou,

Thank you for submitting your manuscript to PLOS ONE. After careful consideration, we feel that it has merit but does not fully meet PLOS ONE’s publication criteria as it currently stands. Therefore, we invite you to submit a revised version of the manuscript that addresses the points raised during the review process.

Please make major revisions based on the reviewers' views. Especially some of the problems pointed out by the third reviewer.

We look forward to receiving your revised manuscript.

Kind regards,

De-Chih Lee, Ph.D.

Academic Editor

PLOS ONE

Reviewers' comments:

Reviewer's Responses to Questions

**Comments to the Author**

1. Is the manuscript technically sound, and do the data support the conclusions?

Reviewer #1: Partly

Reviewer #2: Yes

Reviewer #3: Partly

2. Has the statistical analysis been performed appropriately and rigorously? 

Reviewer #1: No

Reviewer #2: Yes

Reviewer #3: No

3. Have the authors made all data underlying the findings in their manuscript fully available?

Reviewer #1: Yes

Reviewer #2: No

Reviewer #3: Yes

4. Is the manuscript presented in an intelligible fashion and written in standard English?

Reviewer #1: Yes

Reviewer #2: Yes

Reviewer #3: Yes

5. Review Comments to the Author

Reviewer #1: This study is about the factors of commercial medical insurance enrollment with the CGSS data. This is a study with big sample size, and results are reliable. However, improvements still be needed before publication. Here are some recommendations and suggestions for authors to consider.

1. in the introduction part, the knowledge gaps should be mentioned based on previous studies.

2. in the results part, multivariate logistical regression is recommended for factors analysis

3. in the discussion part, many times the medical risk or health risk were mentioned. But I think medical insurance helps people deal with financial risk. So, medical/health risk is not appropriate. Besides, it should be better to discuss the strengths of this study.

Reviewer #2: The research topic is of great reference value. In the future, data from neighboring countries can be added for comparison. Since only 12% of the overall data is registered with CMI, more attention should be paid to the proportion when adding samples for analysis in the future.

Reviewer #3: China has a vast territory and a large population, and promoting commercial medical insurance (CMI) can indeed play a certain role in solving disease burden, health inequalities and other health care challenges.

The author/ authors used Mann-Whitney U test and Chi-square test, and Binary logistic regression analysis to analyze the data from the 2018 Chinese General Social Survey (CGSS), with the purpose of exploring factors related to CMI enrollment, including sociodemographic characteristics, health status, and family economic status. and social security conditions to provide reference for the design and development of China's CMI.

However, the paper suffers from several flaws, some of which appear to be significant, greatly reducing the value of this study. Therefore, I offer the following advice to author/authors.

1. Discussing the factors related to CMI enrollment can provide important policy directions when promoting CMI. There are many relevant published literatures, but I have not seen the author/authors doing a review in this area, which is a pity. It is recommended that author/authors read journal articles published in Journal of Risk and Insurance or Geneva Risk and Insurance Review. Because various studies are progressive in their contributions, the author/authors can consider whether it is necessary to introduce category theory, deductive generation, causal classification, etc., but the author/authors still has to make proof and efforts in this regard.

2. This study used data from the 2018 CGSS. Can the author/authors explain why there is only one year of data? Because the CGSS data has been launched since 2003. If multi-year data were available, the author/authors could conduct a richer analysis and discussion.

3. The author used Mann-Whitney U test and Chi-square test, and Binary logistic regression analysis in the analysis. Mann-Whitney U test and Chi-square test can only analyze whether there are significant differences in some relevant variables between the two groups of CMI-enrollees and CMI non-enrollees. Binary logistic regression analysis analyzes the impact of relevant variables on the decision-making of CMI-enrollees and CMI non-enrollees of the tested samples. I think the author's choice of research methods was relatively simple due to insufficient reading of the research literature, which made this study look like a statistical exercise rather than a research paper. Why did the authors not consider the propensity score matching (PSM)? This method may be more suitable for this study. Likewise, the author/authors can understand the operation and application of this method by reading the research literature.

4. Author/authors must clearly state how the explanatory variables in the analytical model are chosen because this affects the interpretation of the analytical results. Because the data chosen by the author is from China, there may be many obvious differences with data from other countries, such as the start time of the development of commercial insurance, differences in policy regulations, etc. These factors that have not been taken into account will cause the CMI enrollment in China to be different from that in other countries. The author's explanation in this part is insufficient.

6. PLOS authors have the option to publish the peer review history of their article (what does this mean?). If published, this will include your full peer review and any attached files.

Reviewer #1: No

Reviewer #2: No

Reviewer #3: No

---

## [Author Response · Author response to Decision Letter 0]

10 Dec 2023

Dear Editor and reviewers,

Thank you very much for your comments, which have helped to improve the rigor and quality of our manuscript. Please see below point-to-point responses and changes in the revised manuscript (highlighted in yellow).

Reviewer #1: 

This study is about the factors of commercial medical insurance enrollment with the CGSS data. This is a study with big sample size, and results are reliable. However, improvements still be needed before publication. Here are some recommendations and suggestions for authors to consider.

1. in the introduction part, the knowledge gaps should be mentioned based on previous studies.

Response: 

Thanks for your suggestion. We added knowledge gaps in the introduction part. The added sentences are “While international literature is available on the factors associated with enrollment of CMI, similar studies remain limited in the context of China. In general, these studies failed to explore the factors associated with the enrollment of CMI in a comprehensive and systematic way for the general population. Prior studies have focused on specific populations such as middle-aged and elderly people, and migrants, [10, 18] with a smaller sample size [8, 17].”

(Please see pages 5-6 of the revised manuscript, lines 91-94.)

Cite-[8] Ying XH, Hu TW, Ren J, Chen W, Xu K, Huang JH. Demand for private health insurance in Chinese urban areas. Health economics. 2007;16(10):1041-50. Epub 2007/01/03. doi: 10.1002/hec.1206. PubMed PMID: 17199233.

 [10] Liu J, Chen H, Yang C, Li Z. Exploring the relationship between migrants' purchasing of commercial medical insurance and urbanisation in China. BMC health services research. 2018;18(1):679.

[17] Chen M, Chen W, Zhao Y. New evidence on financing equity in China's health care reform--a case study on Gansu province, China. BMC health services research. 2012;12:466. Epub 2012/12/19. doi: 10.1186/1472-6963-12-466. PubMed PMID: 23244513; PubMed Central PMCID: PMCPMC3562140.

[18] Jin Y, Hou Z, Zhang D. Determinants of Health Insurance Coverage among People Aged 45 and over in China: Who Buys Public, Private and Multiple Insurance. PloS one. 2016;11(8):e0161774. Epub 2016/08/27. doi: 10.1371/journal.pone.0161774. PubMed PMID: 27564320; PubMed Central PMCID: PMCPMC5001699.

2. in the results part, multivariate logistical regression is recommended for factors analysis.

Response: 

Statistically speaking, multivariate analyses refer to statistical models with two or more dependent or outcome variables, and multivariable analysis refers to statistical models with multiple independent variables and one dependent variable (citation: Hidalgo B, Goodman M. Multivariate or multivariable regression? American journal of public health. 2013;103(1):39-40. Epub 2012/11/17. doi: 10.2105/ajph.2012.300897. PubMed PMID: 23153131; PubMed Central PMCID: PMCPMC3518362.). Based on the research objective, we only have one dependent variable, so multivariable logistical regression is adopted. This approach is widely used for such a type of analysis.

3. in the discussion part, many times the medical risk or health risk were mentioned. But I think medical insurance helps people deal with financial risk. So, medical/health risk is not appropriate. Besides, it should be better to discuss the strengths of this study.

Response：

Thanks for your suggestions. In the discussion part, we have changed “medical risk or health risk” to “financial risks brought by medical problems” wherever relevant.

We add two strengths of this study and add them to the revised manuscript: “The main strength of this study lies in the large national representative sample(n=11,666), which helps to improve the validity of our conclusions. In addition, our study provides comprehensive and systematic factors associated with the enrollment of CMI covering sociodemographic characteristics, health status, family economic status, and social security status. It would help develop targeted policy suggestions.” 

(Please see page 20-21 of the revised manuscript, lines 345-349.)

Reviewer #2: 

The research topic is of great reference value. In the future, data from neighboring countries can be added for comparison. Since only 12% of the overall data is registered with CMI, more attention should be paid to the proportion when adding samples for analysis in the future.

Response: 

Thanks for your positive comments. Based on the comment, we have added some discussions about the difference in coverage between China and other countries such as the United States although they are not neighboring countries. The choice of comparing China to those countries is because it helps readers understand other potential factors affecting the enrollment rate (beyond those generated from the multivariable analysis), such as policies and development time of CMI. 

Reviewer #3: 

China has a vast territory and a large population, and promoting commercial medical insurance (CMI) can indeed play a certain role in solving disease burden, health inequalities and other health care challenges.

The author/ authors used Mann-Whitney U test and Chi-square test, and Binary logistic regression analysis to analyze the data from the 2018 Chinese General Social Survey (CGSS), with the purpose of exploring factors related to CMI enrollment, including sociodemographic characteristics, health status, and family economic status. and social security conditions to provide reference for the design and development of China's CMI. 

However, the paper suffers from several flaws, some of which appear to be significant, greatly reducing the value of this study. Therefore, I offer the following advice to author/authors.

1.Discussing the factors related to CMI enrollment can provide important policy directions when promoting CMI. There are many relevant published literatures, but I have not seen the author/authors doing a review in this area, which is a pity. It is recommended that author/authors read journal articles published in Journal of Risk and Insurance or Geneva Risk and Insurance Review. Because various studies are progressive in their contributions, the author/authors can consider whether it is necessary to introduce category theory, deductive generation, causal classification, etc., but the author/authors still has to make proof and efforts in this regard.

Response：

Thanks for the comments. This study, as an exploratory one in China, aims to explore whether the factors that were studied before in other settings are associated with the enrollment of CMI in China. The factors included in the regression model were primarily based on the literature review, which was mostly referenced in the third paragraph of the background section. Based on the reviewer’s comment and to make it clearer, we added the relevant articles published in the Journal of Risk and Insurance or Geneva Risk and Insurance Review to enrich the review content. The added articles include: “Brown JR, Finkelstein A. The Private Market for Long-Term Care Insurance in the U.S.: A Review of the Evidence. The Journal of risk and insurance. 2009;76(1):5-29. Epub 2010/01/05. doi: 10.1111/j.1539-6975.2009.01286.x. PubMed PMID: 20046809; PubMed Central PMCID: PMCPMC2799900.”,“Harris TF, Yelowitz A, Courtemanche C. Did COVID-19 change life insurance offerings? The Journal of risk and insurance. 2021;88(4):831-61. Epub 2021/07/07. doi: 10.1111/jori.12344. PubMed PMID: 34226761; PubMed Central PMCID: PMCPMC8242708.”,“ Gollier C. Insurance economics and COVID-19. The Journal of risk and insurance. 2021;88(4):825-9. Epub 2021/12/16. doi: 10.1111/jori.12366. PubMed PMID: 34908588; PubMed Central PMCID: PMCPMC8662029.”

We are grateful about the suggestions by the reviewer to consider those theories. While category theory is the most general and abstract branch of pure mathematics, deductive generation is helpful to testify the theory and hypothesis, and causal classification aims to discover causal relationships between variables and understand the causal mechanisms behind the relationship, we feel the conventional approach is sufficient to address the research question, since this is an exploratory study which aims to explore the factors associated with the enrollment of CMI in China. As we are mainly from the areas of public health and health system, we have proposed a more applied research question. Yet in the future, we could attempt to apply these and other economic theories to improve the theoretical depth of the insurance studies. 

2.This study used data from the 2018 CGSS. Can the author/authors explain why there is only one year of data? Because the CGSS data has been launched since 2003. If multi-year data were available, the author/authors could conduct a richer analysis and discussion.

Response：

We appreciate the reviewer’s comment. In the last decades, the enrollment of public health insurance has expanded substantially. At the same time, Chinese society has undergone drastic changes, such as urbanization and education expansion. With drastic societal changes, the same measure in 2003 could be quite different in 2018. Taking education as an example, a person with a college degree in 2003 was viewed quite differently from a person with a college degree in 2018. The same is true for the measurement of employment (e.g. farming). Given the difference in what is measured by the same variable and the main purpose of this study to inform further policy, this study used the latest available data, which is CGSS in 2018, to perform the analysis. CGSS 2018 contains rich data covering 11,666 individuals and provides a good foundation for the exploration of the factors influencing the enrollment of commercial health insurance in China. We acknowledged that potential panel data analysis could be considered for future studies. The specific description is “Second, we have adopted a cross-sectional study using the latest data from the 2018 CGSS, the most recent data published by CGSS at the time of writing. Future studies could adopt a longitudinal design to explore the relationship between CMI enrollment and associated factors. ”

(Please see page 21 of the revised manuscript, lines 352-355.)

3.The author used Mann-Whitney U test and Chi-square test, and Binary logistic regression analysis in the analysis. Mann-Whitney U test and Chi-square test can only analyze whether there are significant differences in some relevant variables between the two groups of CMI-enrollees and CMI non-enrollees. Binary logistic regression analysis analyzes the impact of relevant variables on the decision making of CMI-enrollees and CMI non-enrollees of the tested samples. I think the author's choice of research methods was relatively simple due to insufficient reading of the research literature, which made this study look like a statistical exercise rather than a research paper. Why did the authors not consider the propensity score matching (PSM)? This method may be more suitable for this study. Likewise, the author/authors can understand the operation and application of this method by reading the research literature. 

Response：

We appreciate the reviewer's suggestion of PSM. We chose logistic regression for two major reasons. Firstly, logistic regression is a valid approach and has been widely used to examine the relationship between health insurance enrollment and potential factors (The references include:“Nguyen TD, Wilson A. Coverage of health insurance among the near-poor in rural Vietnam and associated factors. International journal of public health. 2017;62(Suppl 1):63-73. Epub 2016/11/01. doi: 10.1007/s00038-016-0911-z. PubMed PMID: 27796412.”, “Kazungu JS, Barasa EW. Examining levels, distribution and correlates of health insurance coverage in Kenya. Tropical medicine & international health : TM & IH. 2017;22(9):1175-85. Epub 2017/06/20. doi: 10.1111/tmi.12912. PubMed PMID: 28627085; PubMed Central PMCID: PMCPMC5599961.”,“Tan C, Wyatt LC, Kranick JA, Kwon SC, Oyebode O. Factors Associated with Health Insurance Status in an Asian American Population in New York City: Analysis of a Community-Based Survey. Journal of racial and ethnic health disparities. 2018;5(6):1354-64. Epub 2018/03/28. doi: 10.1007/s40615-018-0485-y. PubMed PMID: 29582383; PubMed Central PMCID: PMCPMC6158120.”). Using the same methodology as the literature would allow for better comparisons of the results. Secondly, in our sample, we have a relatively large number of enrolled individuals (1379) and more than 10 co-variates. A study shows that “in terms of the tradeoff between bias and precision, once there are eight events per confounder, logistic regression is a better approach.”(The reference: “Cepeda MS, Boston R, Farrar JT, Strom BL. Comparison of logistic regression versus propensity score when the number of events is low and there are multiple confounders. American journal of epidemiology. 2003;158(3):280-7. Epub 2003/07/29. doi: 10.1093/aje/kwg115. PubMed PMID: 12882951.”) In this study, we have a ratio of more than 100, and logistic regression would perform at least as well as PSM, in our opinion. 

The importance of this study was described in the introduction section. Given the high out-of-pocket health spending in China, understanding the factors associated with supplemental health insurance remains to be an important policy question.

We hope these explanations mitigate the reviewer’s concern.

4.Author/authors must clearly state how the explanatory variables in the analytical model are chosen because this affects the interpretation of the analytical results. Because the data chosen by the author is from China, there may be many obvious differences with data from other countries, such as the start time of the development of commercial insurance, differences in policy regulations, etc. These factors that have not been taken into account will cause the CMI enrollment in China to be different from that in other countries. The author's explanation in this part is insufficient.

Response：

Thanks for the comment. This is an exploratory study. The choice of explanatory variable was primarily based on relevant published journal articles. This paper selects the data from eight sections of the CGSS questionnaire, including (a) sociodemographic characteristics (e.g. age, gender, education), (c) health status (e.g. self-assessed physical health), (d) migration status (e.g. household registration status), (e) lifestyle behaviors, (g) class identity, (j) participation of labor market, (k) social security status, and (l) family characteristics (e.g. household size). These selected variables were further sorted into four categories: sociodemographic characteristics (age, sex, marital status, education, number of family members, current work experience and status, household registration, subjective social status), health status (BMI, self-assessment of physical health, hospitalization, frequency of physical exercise), family economic status ( house, car, investment, level of family economy ), and social security status (participate in basic medical insurance, participate in basic pension insurance). All these categories of variables were widely used in other countries and settings. 

We acknowledge that the impact of identified factors on enrollment of private health insurance could be different between China and other countries given the context difference mentioned by the reviewer. To address the concern, the discussion of the factors such as the start time of the development of commercial insurance and differences in policy regulations is now added to the discussion section. The CMI of the United States is used as an example to analyze the differences between China and the United States in CMI. The specific description is “However, compared with the United States and some developed European countries, the coverage of CMI in China is relatively low. The differences in coverage may be influenced by factors such as the start time of the development of CMI and policy regulations. For instance, in the United States where the “market-oriented” health insurance model is dominant, 55.3% of individuals were enrolled in CMI in 2016 [34]. The high coverage may be related to the earlier establishment and development of the CMI system in the United States which began around

---

## [Decision Letter · Decision Letter 1]

2 Jan 2024

PONE-D-23-25788R1Factors associated with the enrollment of commercial medical insurance in China: results from China General Social SurveyPLOS ONE

Dear Dr. Zou,

Thank you for submitting your manuscript to PLOS ONE. After careful consideration, we feel that it has merit but does not fully meet PLOS ONE’s publication criteria as it currently stands. Therefore, we invite you to submit a revised version of the manuscript that addresses the points raised during the review process.

We look forward to receiving your revised manuscript.

Kind regards,

De-Chih Lee, Ph.D.

Academic Editor

PLOS ONE

**Additional Editor Comments:**

Please make major revisions based on the comments of the two reviewers.

Reviewers' comments:

Reviewer's Responses to Questions

**Comments to the Author**

1. If the authors have adequately addressed your comments raised in a previous round of review and you feel that this manuscript is now acceptable for publication, you may indicate that here to bypass the “Comments to the Author” section, enter your conflict of interest statement in the “Confidential to Editor” section, and submit your "Accept" recommendation.

Reviewer #1: (No Response)

Reviewer #3: All comments have been addressed

2. Is the manuscript technically sound, and do the data support the conclusions?

Reviewer #1: Partly

Reviewer #3: Partly

3. Has the statistical analysis been performed appropriately and rigorously? 

Reviewer #1: No

Reviewer #3: No

4. Have the authors made all data underlying the findings in their manuscript fully available?

Reviewer #1: (No Response)

Reviewer #3: No

5. Is the manuscript presented in an intelligible fashion and written in standard English?

Reviewer #1: (No Response)

Reviewer #3: Yes

6. Review Comments to the Author

Reviewer #1: The revised paper was improved a lot.

About the multivariate logistical regression issue I mentioned last time, I appreciated that that author provided me with a reference about the difference between multivariate/multivariable regression. What I wanted to say last time is that multivariable regression should be taken. In the author’s response, they claimed that ‘multivariable regression’ was adopted. However, in the methods part, “multivariable analysis” was used to describe the groups comparisons but not logistic regression. Authors may misunderstand the term “multivariable analysis”, which does not mean that multiple variables are analyzed independently but are analyzed simultaneously to get combined results for each variable. For instance, when we analyze marital status and enrollment in table 5, we put age and/or other variables in. That is multivariable regression, which can be used to adjust the relationship between one independent variable and enrollment by introducing other variables in analysis.

A multivariable regression analysis is recommended for consideration again.

Reviewer #3: Thanks to the editor-in-chief and the author for giving me the opportunity to read this paper again. I can clearly feel the author's efforts and the improvement in the quality of the paper. However, I still have some concerns about the current version of the paper and would like to provide the author with suggestions for revision.

1. Regarding the first suggestion I made after the previous reading, I hope the author will introduce necessary category theory, deductive generation, causal classification, etc., because even though this article is an exploratory study, I think many analysis variables are set and proposed. It still requires theoretical basis or inference, and the author's improvement in this part is still insufficient.

2. I still hope that the author will increase the number of data years analyzed instead of only using the data of 2018 for a cross-sectional analysis. I agree that the environmental background of the 2003 data and the 2018 data is very different, but the author can use the data from 5 or 8 years before 2018 and then use the panel model. I would like to emphasize that the conclusions of many studies may be the result of institutional or environmental changes over the years. Therefore, I hope that the authors will consider using multi-year data again on the premise of data availability.

3. I asked the author to review studies in other countries similar to this paper because I hope the author can explore the possible reasons for the differences between China and other countries on some variables. There must be certain social, economic or institutional variables that cause the enrollment of commercial medical insurance in China to be different from that in other countries. Therefore, I suggest that the author still needs to review other studies and provide explanations. Alternatively, the suggestion I made in the first point is also a direction that the author can take. The purpose of doing so is to improve the academic contribution of this paper.

7. PLOS authors have the option to publish the peer review history of their article (what does this mean?). If published, this will include your full peer review and any attached files.

Reviewer #1: No

Reviewer #3: No

---

## [Author Response · Author response to Decision Letter 1]

28 Mar 2024

Dear Editor and reviewers,

 Thank you very much for your comments, which have helped to improve the rigor and quality of our manuscript. Please see below point-to-point responses and changes in the revised manuscript (highlighted in yellow).

 Reviewer #1: 

 The revised paper was improved a lot. About the multivariate logistical regression issue I mentioned last time, I appreciated that that author provided me with a reference about the difference between multivariate/multivariable regression. What I wanted to say last time is that multivariable regression should be taken. In the author’s response, they claimed that “multivariable regression” was adopted. However, in the methods part, “multivariable analysis” was used to describe the groups comparisons but not logistic regression. Authors may misunderstand the term “multivariable analysis”, which does not mean that multiple variables are analyzed independently but are analyzed simultaneously to get combined results for each variable. For instance, when we analyze marital status and enrollment in table 5, we put age and/or other variables in. That is multivariable regression, which can be used to adjust the relationship between one independent variable and enrollment by introducing other variables in analysis. A multivariable regression analysis is recommended for consideration again.

 Response: 

 Thanks for the clarification. We used logistic regression models to perform the multivariable analysis. The independent variables included in the logistic regression model are those that meet certain criteria in the bivariate analysis (e.g. Chi-square test and Mann-Whitney U test). The independent variables in the logistic regression were analyzed simultaneously instead of independently. The results are presented in Table 5. We also revised the method section regarding the logistic regression to further clarify it. The revised sentences are “To explore the association of potential factors associated with enrollment of CMI, a multivariable logistic regression model was used. The dependent variable was whether to enroll in CMI (Yes=1, No=0). In this revised submission, the independent variables were explicit characteristics (sociodemographic characteristics and family economic status), latent characteristics (social security status), and the global incentive compatibility index (health status). The independent variables with a p-value less than 0.2 in the Mann-Whitney U test and Chi-square test were included in the subsequent binary logistic regression analysis. The results were presented as adjusted ORs with 95% confidence interval (CI). The significant level was set at 5%”

 (Please see pages 9-10 of the revised manuscript, lines 155-161.)

 Reviewer #3: 

 Thanks to the editor-in-chief and the author for giving me the opportunity to read this paper again. I can clearly feel the author's efforts and the improvement in the quality of the paper. However, I still have some concerns about the current version of the paper and would like to provide the author with suggestions for revision.

 1. Regarding the first suggestion I made after the previous reading, I hope the author will introduce necessary category theory, deductive generation, causal classification, etc., because even though this article is an exploratory study, I think many analysis variables are set and proposed. It still requires theoretical basis or inference, and the author's improvement in this part is still insufficient. 

 Response：

 We appreciate the reviewer’s comments. We introduce the principal-agent model to understand the factors associated with the enrollment of the CMI in China in this revised version. Based on Nganje’s framework on determinants of health insurance according to the idea of the principal-agent model (citation: Nganje W, Addey KA. Health Uninsurance in rural America: a partial equilibrium analysis. Health economics review. 2019;9(1):19. Epub 2019/06/21. doi: 10.1186/s13561-019-0234-x. PubMed PMID: 31218435; PubMed Central PMCID: PMCPMC6734485.), we reviewed the literature on factors associated with health insurance and found that the main potential determinants of health insurance enrollment were primarily categorized into (1) Explicit characteristics, including sociodemographic characteristics(e.g., age, gender, employment, and education) and family economic status(e.g., family size, income, and property); (2) Latent characteristics, such as the existence of competing risk protection mechanisms, can reflect the psychological characteristics of risk-averse (e.g., basic medical insurance); and (3) The global incentive compatibility index, including health-related factors (e.g., lifestyle, and past health experience). 

 2. I still hope that the author will increase the number of data years analyzed instead of only using the data of 2018 for a cross-sectional analysis. I agree that the environmental background of the 2003 data and the 2018 data is very different, but the author can use the data from 5 or 8 years before 2018 and then use the panel model. I would like to emphasize that the conclusions of many studies may be the result of institutional or environmental changes over the years. Therefore, I hope that the authors will consider using multi-year data again on the premise of data availability. 

 Response：

 Thanks for your suggestions. We searched the public data from 5 or 8 years around 2018 on the official website of China General Social Survey (CGSS) and found that data for 2013, 2015, 2017, 2018, and 2021 were available. Considering the availability and consistency of data available for certain variables, and to reflect the patterns of factors associated with enrolment of CMI in China, we chose data for 2015, 2018 and 2021. After removing the missing data, we included 10,305, 11,666 and 7,252 individuals in 2015, 2018, and 2021. We explain this in the “Data sources” section of this article, and the specific description is “We used CGSS in 2015, 2018, and 2021 that contained respondents from 29 provinces/cities/autonomous regions in China. After removing missing data, the datasets in 2015, 2018, and 2021 contain 10,305, 11,666, and 7,252 samples, respectively.”

 (Please see page 8 of the revised manuscript, lines 123-126). 

 After data analysis, the data analysis results of 2015 and 2021 were added to the tables. We found that of all individuals, 8.7%,11.8% and 14.1% had enrolled in CMI in 2015,2018 and 2021, respectively, and the factors associated with CMI were consistent in these three years. Moreover, we found that individuals without basic pension insurance were less likely to enroll in CMI when analyzing the 2015 data. And we also found that obese individuals and divorced individuals were more likely to enroll in CMI when analyzing data for 2021. These findings were added to the “Results” and “Discussion” sections. 

The specific description is as follows.

 “We found that of all individuals, the proportion of enrolled individuals shows an increasing trend year by year, with 8.7%,11.8% and 14.1% enrolled in CMI in 2015,2018 and 2021, respectively.”

 (Please see page 38 of the revised manuscript, lines 272-274).

 “Divorced individuals were 1.48 times more likely to enroll in CMI than married individuals (OR=1.48, 95% CI: 1.03-2.13 in 2021).” “Individuals without basic pension insurance were less likely to enroll in CMI compared to those with basic pension insurance (OR=0.80, 95% CI:0.66-0.97 in 2015).” “In perspective to the global incentive compatibility, obese individuals were 1.20 times more likely to enroll in CMI than those BMI within normal range according to the BMI range (OR=1.20, 95% CI: 1.03-1.39 in 2021).”

 (Please see pages 30-31 of the revised manuscript, lines 241-242, lines 259-261, lines 262-264). 

 “However, we also found that divorced individuals were more likely to enroll in CMI than those who were married, which is different from previous study [11]. This may be because divorced individuals who have been through marriage are more convinced of the importance of medical insurance.”

(Please see page 40 of the revised manuscript, lines 316-319).

Cite-[11] - Balqis-Ali NZ, Anis-Syakira J, Weng HF, Sararaks S. Private Health Insurance in Malaysia: Who Is Left Behind? Asia-Pacific Journal of Public Health. 2021;33(8):861-9.

 “We also found that obese individuals were more likely to enroll in CMI than those BMI within normal range. As obese individuals have a much higher risk of disease than normal individuals, it may be that obese individuals feel they need CMI to reduce the financial risk of medical treatment.”

 (Please see pages 42-43 of the revised manuscript, lines 374-377).

 “However, we found that individuals without basic pension insurance were less likely to enroll in CMI compared to those with basic pension insurance. This may be because individuals without basic pension insurance do not pay attention to the role of insurance, and will not spend more money to enroll in CMI.”

(Please see page 42 of the revised manuscript, lines 363-366).

 3. I asked the author to review studies in other countries similar to this paper because I hope the author can explore the possible reasons for the differences between China and other countries on some variables. There must be certain social, economic or institutional variables that cause the enrollment of commercial medical insurance in China to be different from that in other countries. Therefore, I suggest that the author still needs to review other studies and provide explanations. Alternatively, the suggestion I made in the first point is also a direction that the author can take. The purpose of doing so is to improve the academic contribution of this paper. 

 Response：

 According to your suggestion, we have introduced the principal-agent model in this revised version to understand the factors associated with the enrollment of the CMI in China and analyzed the multi-year data to improve the academic contribution of this paper. 

 Indeed, there must be certain social, economic or institutional variables that cause the enrollment of CMI in China to be different from that in other countries. We did not analyze these factors as variables because there was no available data on these aspects in the questionnaire we used. However, we added some comparison with other countries in these aspects in the previous manuscript, such as the start time of the development of commercial insurance and differences in policy regulations. The specific description is “However, compared with the United States and some developed European countries, the coverage of CMI in China is relatively low. The differences in coverage may be influenced by factors such as the start time of the development of CMI and policy regulations. For instance, in the United States where the “market-oriented” health insurance model is dominant, 55.3% of individuals were enrolled in CMI in 2016 [34]. The high coverage may be related to the earlier establishment and development of the CMI system in the United States which began around 1945 as a pillar of its medical security system [35], and the strong legal protection for those residents who do not have employer-provided health insurance and do not qualify for public insurance to purchase CMI [36]. Besides, the governments of England, Germany, and the United States have all introduced tax support policies for CMI, which has a positive impact on increasing the CMI coverage.”

 (Please see page 38 of the revised manuscript, lines 275-284.)

 Cite-[34] - Papanicolas I, Woskie LR, Jha AK. Health Care Spending in the United States and Other High-Income Countries. Jama. 2018;319(10):1024-39. Epub 2018/03/15. doi: 10.1001/jama.2018.1150. PubMed PMID: 29536101. 

[35] Rice T, Rosenau P, Unruh LY, Barnes AJ. United States: Health System Review. Health systems in transition. 2020;22(4):1-441. Epub 2021/02/03. PubMed PMID: 33527901. 

[36] McIntyre A, Song Z. The US Affordable Care Act: Reflections and directions at the close of a decade. PLoS medicine. 2019;16(2):e1002752. Epub 2019/02/27. doi: 10.1371/journal.pmed.1002752. PubMed PMID: 30807584; PubMed Central PMCID: PMCPMC6390990 following competing interests: ZS is a member of the Editorial Board of PLOS Medicine.

---

## [Decision Letter · Decision Letter 2]

6 May 2024

Factors associated with the enrollment of commercial medical insurance in China: results from China General Social Survey

PONE-D-23-25788R2

Dear Dr. Zou,

We’re pleased to inform you that your manuscript has been judged scientifically suitable for publication and will be formally accepted for publication once it meets all outstanding technical requirements.

Kind regards,

De-Chih Lee, Ph.D.

Academic Editor

PLOS ONE

Additional Editor Comments (optional):

Reviewers' comments:

Reviewer's Responses to Questions

**Comments to the Author**

1. If the authors have adequately addressed your comments raised in a previous round of review and you feel that this manuscript is now acceptable for publication, you may indicate that here to bypass the “Comments to the Author” section, enter your conflict of interest statement in the “Confidential to Editor” section, and submit your "Accept" recommendation.

Reviewer #1: All comments have been addressed

Reviewer #3: All comments have been addressed

2. Is the manuscript technically sound, and do the data support the conclusions?

Reviewer #1: Yes

Reviewer #3: Yes

3. Has the statistical analysis been performed appropriately and rigorously? 

Reviewer #1: Yes

Reviewer #3: Yes

4. Have the authors made all data underlying the findings in their manuscript fully available?

Reviewer #1: Yes

Reviewer #3: Yes

5. Is the manuscript presented in an intelligible fashion and written in standard English?

Reviewer #1: Yes

Reviewer #3: Yes

6. Review Comments to the Author

Reviewer #1: The authors clarified the multivariable regression analysis in the methods part. It is better to add methods used to the title of table 5 or explain the ORs with the methods in captions of table 5. No more concerns arise from this revised paper.

Reviewer #3: I would like to thank the editor-in-chief and the authors again for giving me the opportunity to review this study. The current version has been greatly improved and I will recommend it to the editor-in-chief for publication.

1. The author has added consideration to factors that have been discussed in other literature that affect people's purchase of health insurance, which makes the model more complete.

2. The author has added data from 2015 and 2021, which adds discussion of differences and similarities in different years during the analysis process, allowing readers to understand the changes in factors affecting Chinese people’s health insurance purchase over time.

3. In the end, the author still did not do any comparison of factors affecting health insurance coverage between China and other countries, which is a pity. However, the content of this current version is more complete and richer than the previous two versions. Perhaps the comparison of factors affecting health insurance enrollment in China and other countries can be used as a direction for future extended research.

7. PLOS authors have the option to publish the peer review history of their article (what does this mean?). If published, this will include your full peer review and any attached files.

Reviewer #1: No

Reviewer #3: No

---

## [Editor Report · Acceptance letter]

14 May 2024

PONE-D-23-25788R2 

PLOS ONE

Dear Dr. Zou, 

I'm pleased to inform you that your manuscript has been deemed suitable for publication in PLOS ONE. Congratulations! Your manuscript is now being handed over to our production team.

Kind regards, 

on behalf of

Dr. De-Chih Lee 

Academic Editor

PLOS ONE